# Second-Order Parametric Free Dualities for Complex Minimax Fractional Programming

## Tone-Yau Huang

Department of Applied Mathematics, Feng-Chia University, Tai-Chung 40724, Taiwan; huangty@fcu.edu.tw

**Abstract:** In this paper, we will consider a minimax fractional programming in complex spaces. Since a duality model in a programming problem plays an important role, we will establish the second-order Mond–Weir type and Wolfe type dual models, and derive the weak, strong, and strictly converse duality theorems.

**Keywords:** minimax fractional programming; duality theorems; generalized convexity

---

## 1. Introduction

The minimax theorems are very important results in fixed point theory, game theory, minimax programming problems, etc. John Nash provided an alternative proof of the minimax theorem using Brouwer's fixed point theorem. Later, Fan [1] established some minimax theorems on nonlinear spaces. For instance, if $X, Y$ are the compact Hausdorff spaces, and if the real-valued functional $f : X \times Y \to \mathbb{R}$ satisfies some suitable conditions, then

$$\min_{\mathbf{x} \in X} \max_{\mathbf{y} \in Y} f(\mathbf{x}, \mathbf{y}) = \max_{\mathbf{y} \in Y} \min_{\mathbf{x} \in X} f(\mathbf{x}, \mathbf{y}).$$

On the other hand, many authors considered the left-hand side of the above equality with some constraints, as a minimax programming problem. In 1977, Schmittendorf [2] first considered the following real minimax problem:

$$\min_{\mathbf{x} \in X} \sup_{\mathbf{y} \in Y} \quad f(\mathbf{x}, \mathbf{y})$$
$$\text{s.t.} \qquad X = \{ \mathbf{x} \in \mathbb{R}^n \,|\, h(\mathbf{x}) < 0 \},$$

where $Y$ is a compact subset in $\mathbb{R}^m$, functions $f : \mathbb{R}^n \times \mathbb{R}^m \to \mathbb{R}$ and $h : \mathbb{R}^n \to \mathbb{R}^p$ are $\mathcal{C}^1$ mappings. Since then, authors were interested in various types of the real minimax programming problems. They derived the necessary and sufficient optimality conditions, and investigated various types of duality models (see [3–7]). The generalized convexity is an important mathematical tool for studying the sufficient optimality conditions and duality models in programming problems. For instance, Mititelu and Treanţă [8] studied the efficiency conditions in vector control problems governed by multiple integrals. Treanţă and Mititelu [9] investigated the duality with $(\rho, b)$-quasiinvexity for multidimensional vector fractional control problems. Cipu [10] considered the duality results in quasiinvex variational control problems with curvilinear integral functionals. Treanţă [11] also studied the multiobjective fractional variational problem on higher-order jet bundles.

For the minimax programming problems in complex spaces, Datta and Bhatia [12] first considered the following complex minimax programming:

$$\min_{\mathbf{x}\in X}\ \sup_{\mathbf{y}\in Y}\ \operatorname{Re} f(\mathbf{x},\mathbf{y})$$
$$\text{s.t.}\qquad \mathbf{x}\in X = \{\mathbf{x}\in \mathbb{C}^{2n}\mid -h(\mathbf{x})\in T\},$$

where $Y$ is a compact subset in $\mathbb{C}^{2m}$, $T$ is a polyhedral cone in $\mathbb{C}^p$, functions $f(\cdot,\mathbf{y})$ and $h(\cdot)$ are analytic on $Q = \{\mathbf{x} = (x,\bar{x})\mid x\in\mathbb{C}^n\}\subset \mathbb{C}^{2n}$. Remark that a nonlinear analytic function $f:\mathbb{C}^n\to\mathbb{C}$ does not have a convex real part. As a consequence, we consider the complex functions defined on a linear manifold of the set $Q = \{\mathbf{x} = (x,\bar{x})\mid x\in\mathbb{C}^n\}\subset\mathbb{C}^{2n}$ (see Ferrero [13]).

After then, authors considered various types of the complex minimax programming problems, established the optimality conditions, and studied various types of duality models under some generalized convexities (see [14–18]).

In 2017, Huang [19] has constructed the second-order duality for non-differentiable complex minimax programming problems. Huang and Lai [20] also established the second-order parametric duality for complex minimax fractional programming problem, and derived the duality theorems under generalized $\Theta$-bonvexity.

In this paper, we are interested in a complex fractional minimax programming problem:

$$\text{(P)}\quad \min_{\mathbf{x}\in X}\ \sup_{\mathbf{y}\in Y}\ \frac{\operatorname{Re} f(\mathbf{x},\mathbf{y})}{\operatorname{Re} g(\mathbf{x},\mathbf{y})}$$
$$\text{subject to}\quad \mathbf{x}\in X = \{\mathbf{x}\in\mathbb{C}^{2n}\mid -h(\mathbf{x})\in T\},$$

where $Y$ is a specified compact subset in $\mathbb{C}^{2m}$, $T$ is a polyhedral cone in $\mathbb{C}^p$; for $\mathbf{x} = (x,\bar{x})\in\mathbb{C}^{2n}$, $\mathbf{y} = (y,\overline{y})\in\mathbb{C}^{2m}$, functions $f(\cdot,\cdot)$ and $g(\cdot,\cdot)$ are continuous functions; for each $\mathbf{y}\in Y$, $f(\cdot,\mathbf{y})$, $g(\cdot,\mathbf{y})$ and $h(\cdot)$ are analytic on the $Q = \{\mathbf{x} = (x,\bar{x})\mid x\in\mathbb{C}^n\}\subset\mathbb{C}^{2n}$, and, without loss of generality, we could assume that $\operatorname{Re} f(\mathbf{x},\mathbf{y})\geq 0$ and $\operatorname{Re} g(\mathbf{x},\mathbf{y}) > 0$.

Our main goals of this paper will establish two types of second-order parametric free dual model for the complex minimax fractional programming problem (P), and prove that the weak, strong, and strictly converse duality theorems under generalized $\Theta$-bonvexity assumptions.

This paper is divided into five sections. In order to construct the second-order parametric free dual models, the definition of the second-order $\Theta$-bonvexity, some notations and lemmas are introduced in Section 2. Sections 3 and 4 include main results in this paper: we will formulate the second-order Mond–Weir type (2nd-MWD) and Wolfe type (2nd-WD) dual models for problem (P) using the necessary optimality conditions theorem with some suitable constraints, and derive their duality theorems under generalized $\Theta$-bonvexity. In Section 5, we will talk about the further plausible work.

## 2. Preliminary

In this section, we recall notations, definitions and introduce some lemmas from [15,19]. Given $\mathbf{x}\in\mathbb{C}^p$, the notations $\bar{\mathbf{x}}$, $\mathbf{x}^T$ and $\mathbf{x}^H$ are conjugate, transpose, and transpose conjugate of $\mathbf{x}$. Let $M\in\mathbb{C}^{k\times p}$ be a $k\times p$ matrix, and the set $T = \{\mathbf{x}\in\mathbb{C}^p\mid\operatorname{Re}(M\mathbf{x})\geq 0\}$ be a polyhedral cone. The dual (or polar) cone $T^*$ of $T$ is defined by

$$T^* = \{\mu\in\mathbb{C}^p\mid\operatorname{Re}\langle\mathbf{x},\mu\rangle\geq 0\text{ for }\mathbf{x}\in T\},$$

where $\langle\mathbf{x},\mu\rangle = \mu^H x$ is the inner product in complex spaces. Remark that $T = (T^*)^*$ if $T$ is a polyhedral cone.

Given $\mathbf{x} = (x,\bar{x})\in\mathbb{C}^{2n}$ and a twice differentiable analytic function $\Phi:\mathbb{C}^{2n}\to\mathbb{C}$, the gradient expression $\nabla\Phi(\mathbf{x})$ is denoted by

$$\nabla\Phi(\mathbf{x}) = \big(\nabla_x\Phi(\mathbf{x}),\nabla_{\bar{x}}\Phi(\mathbf{x})\big)\in\mathbb{C}^{2n},$$

where $\nabla_x \Phi(\mathbf{x}) = \left(\Phi_{x_1}(\mathbf{x}), \cdots, \Phi_{x_n}(\mathbf{x})\right) \in \mathbb{C}^n$, $\nabla_{\overline{x}} \Phi(\mathbf{x}) = \left(\Phi_{\overline{x_1}}(\mathbf{x}), \cdots, \Phi_{\overline{x_n}}(\mathbf{x})\right) \in \mathbb{C}^n$.

The second-order gradient expression $\nabla^2 \Phi(\mathbf{x})$ is denoted by

$$\nabla^2 \Phi(\mathbf{x}) = \begin{pmatrix} \nabla_{xx} \Phi(\mathbf{x}), \ \nabla_{x\overline{x}} \Phi(\mathbf{x}) \\ \nabla_{\overline{x}x} \Phi(\mathbf{x}), \ \nabla_{\overline{x}\overline{x}} \Phi(\mathbf{x}) \end{pmatrix} \in \mathbb{C}^{2n \times 2n},$$

with

$\nabla_{xx} \Phi(\mathbf{x}) = \left(\Phi_{x_i x_j}(\mathbf{x})\right)_{n \times n}, i, j = 1, \ldots n$, $\nabla_{\overline{x}x} \Phi(\mathbf{x}) = \left(\Phi_{\overline{x_i} x_j}(\mathbf{x})\right)_{n \times n}, i, j = 1, \ldots n$,

$\nabla_{x\overline{x}} \Phi(\mathbf{x}) = \left(\Phi_{x_i \overline{x_j}}(\mathbf{x})\right)_{n \times n}, i, j = 1, \ldots n$, $\nabla_{\overline{x}\overline{x}} \Phi(\mathbf{x}) = \left(\Phi_{\overline{x_i}\overline{x_j}}(\mathbf{x})\right)_{n \times n}, i, j = 1, \ldots n$.

In order to introduce the optimality conditions and duality models, we need the following lemmas. For the complete proofs, one can refer to the papers: ([15] Lemma 2) and ([19] Lemma 3.1).

**Lemma 1.** *For* $\mathbf{y} \in Y \subset \mathbb{C}^{2m}$, $\mathbf{x} = (x, \overline{x}) \in Q \subset \mathbb{C}^{2n}$ *and nonzero vector* $\mu \in \mathbb{C}^p$. *Suppose that function*

$$\Phi(\mathbf{x}) = f(\mathbf{x}, \mathbf{y}) + \langle h(\mathbf{x}), \mu \rangle$$

*is differentiable at* $\mathbf{x_0} = (x_0, \overline{x_0}) \in Q$. *Then,*

$$\mathrm{Re}[\Phi'(\mathbf{x_0})(\mathbf{x} - \mathbf{x_0})] = \mathrm{Re}\left[\left\langle x - x_0, \ \overline{\nabla_x f(\mathbf{x_0}, \mathbf{y})} + \nabla_{\overline{x}} f(\mathbf{x_0}, \mathbf{y}) + \mu^T \overline{\nabla_x h(\mathbf{x_0})} + \mu^H \nabla_{\overline{x}} h(\mathbf{x_0}) \right\rangle\right].$$

**Lemma 2.** *Given* $\mathbf{x} = (x, \overline{x}), \mathbf{x_0} = (x_0, \overline{x_0}) \in Q \subset \mathbb{C}^{2n}$, *nonzero vector* $\mu \in \mathbb{C}^p$ *and let* $\mathbf{v} = (v, \overline{v}) = \mathbf{x} - \mathbf{x_0}$. *Then, the twice differentiable analytic functions* $f(\mathbf{x})$ *and* $\langle h(\mathbf{x}), \mu \rangle$ *have the second-order gradient representations at* $\mathbf{x_0}$ *as follows:*

*(a)*

$$(\mathbf{x} - \mathbf{x_0})^T \nabla^2 f(\mathbf{x_0})(\mathbf{x} - \mathbf{x_0}) = \left\langle v, v^H \overline{[\nabla_{xx} f(\mathbf{x_0})]} \right\rangle + \left\langle v^H [\nabla_{\overline{x}\overline{x}} f(\mathbf{x_0})], v \right\rangle$$
$$+ \left\langle v, v^T [\nabla_{\overline{x}x} f(\mathbf{x_0})] \right\rangle + \left\langle v^T [\nabla_{x\overline{x}} f(\mathbf{x_0})], v \right\rangle.$$

*The real part of the above identity is equal to*

$$\mathrm{Re}\left(\left\langle v, v^H \left[\overline{\nabla_{xx} f(\mathbf{x_0})} + \nabla_{\overline{x}\overline{x}} f(\mathbf{x_0})\right] + v^T \left[\overline{\nabla_{\overline{x}x} f(\mathbf{x_0})} + \nabla_{x\overline{x}} f(\mathbf{x_0})\right] \right\rangle\right).$$

*(b)*

$$(\mathbf{x} - \mathbf{x_0})^T \nabla^2 \langle h(\mathbf{x_0}), \mu \rangle (\mathbf{x} - \mathbf{x_0}) = \left\langle v, v^H [\mu^T \overline{\nabla_{xx} h(\mathbf{x_0})}] \right\rangle + \left\langle v^H [\mu^H \nabla_{\overline{x}\overline{x}} h(\mathbf{x_0})], v \right\rangle$$
$$+ \left\langle v, v^T [\mu^T \overline{\nabla_{\overline{x}x} h(\mathbf{x_0})}] \right\rangle + \left\langle v^T [\mu^H \nabla_{x\overline{x}} h(\mathbf{x_0})], v \right\rangle.$$

*The real part of the above identity is equal to*

$$\mathrm{Re}\left(\left\langle v, v^H \left[\mu^T \overline{\nabla_{xx} h(\mathbf{x_0})} + \mu^H \nabla_{\overline{x}\overline{x}} h(\mathbf{x_0})\right] + v^T \left[\mu^T \overline{\nabla_{\overline{x}x} h(\mathbf{x_0})} + \mu^H \nabla_{x\overline{x}} h(\mathbf{x_0})\right] \right\rangle\right).$$

Let $\mathbf{x} \in Q \subset \mathbb{C}^{2n}$ be any feasible solution of problem (P). Denote a set

$$Y(\mathbf{x}) = \left\{ \mathbf{y} \in Y \ \middle| \ \frac{\mathrm{Re}\, f(\mathbf{x}, \mathbf{y})}{\mathrm{Re}\, g(\mathbf{x}, \mathbf{y})} = \sup_{v \in Y} \frac{\mathrm{Re}\, f(\mathbf{x}, v)}{\mathrm{Re}\, g(\mathbf{x}, v)} \right\}.$$

Since $f(\mathbf{x}, \cdot)$ and $g(\mathbf{x}, \cdot)$ are continuous on the compact set $Y$, the set $Y(\mathbf{x})$ is also a compact subset of $Y$, and then the objective function of problem (P) can be expressed by the form:

$$\Psi(\mathbf{x}) = \sup_{\mathbf{y} \in Y} \frac{\mathrm{Re}\, f(\mathbf{x}, \mathbf{y})}{\mathrm{Re}\, g(\mathbf{x}, \mathbf{y})} = \frac{\sum\limits_{i=1}^{k} \lambda_i \, \mathrm{Re}\, f(\mathbf{x}, \mathbf{y}_i)}{\sum\limits_{i=1}^{k} \lambda_i \, \mathrm{Re}\, g(\mathbf{x}, \mathbf{y}_i)}, \tag{1}$$

where $\mathbf{y}_i \in Y(\mathbf{x})$ for $i = 1, \ldots, k$, $\lambda_i > 0$, with $\sum_{i=1}^{k} \lambda_i = 1$, and the problem (P) become

$$(\text{P}) \qquad \min_{\mathbf{x} \in X} \Psi(\mathbf{x}).$$

Now, we could recall the necessary optimality conditions theorem of (P) as follows.

**Theorem 1** ([14] Theorem 3.1). *Let* $\mathbf{x}_0 = (x_0, \overline{x_0})$ *be a (P)-optimal with optimal value* $\gamma^*$. *Suppose that the problem (P) satisfies the constraint qualification at* $\mathbf{x}_0$. *Then, there exists a positive integer* $k$, *scalars* $\lambda_i \geq 0$ *with* $\sum_{i=1}^{k} \lambda_i = 1$, *vectors* $\mathbf{y}_i \in Y(\mathbf{x}_0)$ *for* $i = 1, \ldots, k$ *and non-zero vector* $\mu \in T^* \subset \mathbb{C}^p$ *such that*

$$\sum_{i=1}^{k} \lambda_i \left\{ \left[ \overline{\nabla_x f(\mathbf{x}_0, \mathbf{x}_i)} + \nabla_{\overline{x}} f(\mathbf{x}_0, \mathbf{y}_i) \right] - \gamma^* \left[ \overline{\nabla_x g(\mathbf{x}_0, \mathbf{y}_i)} + \nabla_{\overline{x}} g(\mathbf{x}_0, \mathbf{y}_i) \right] \right\} \tag{2}$$

$$+ \mu^T \overline{\nabla_x h(\mathbf{x}_0)} + \mu^H \nabla_{\overline{x}} h(\mathbf{x}_0) = 0,$$

$$\text{Re} \left[ f(\mathbf{x}_0, \mathbf{y}_i) - \gamma^* g(\mathbf{x}_0, \mathbf{y}_i) \right] = 0, \quad i = 1, 2, \cdots, k, \tag{3}$$

$$\text{Re} \langle \mu, h(\mathbf{x}_0) \rangle = 0. \tag{4}$$

Note that problem (P) is said to have constraint qualification at $\mathbf{x}_0$, if, for any nonzero $\mu \in T^* \subset \mathbb{C}^p$, it results $\mu^T \overline{\nabla_x h(\mathbf{x}_0)} + \mu^H \nabla_{\overline{x}} h(\mathbf{x}_0) \neq 0$.

We state the definition of the generalized second-order $\Theta$-bonvexity as follows.

**Definition 1** ([19] Definition 4.1). *The real part of a twice differentiable analytic function* $\Phi(\cdot)$ *from* $\mathbb{C}^{2n}$ *to* $\mathbb{R}$ *is called, respectively,*

(i) **(strictly)** $\Theta$**-bonvex** *at* $\mathbf{x}_0 \in Q \subset \mathbb{C}^{2n}$ *if there exists a certain mapping* $\Theta : \mathbb{C}^{2n} \times \mathbb{C}^{2n} \to \mathbb{C}^{2n}$ *such that for any* $\mathbf{x} \in Q$,

$$\text{Re} \left\{ \Phi(\mathbf{x}) - \Phi(\mathbf{x}_0) + \tfrac{1}{2}(\mathbf{x} - \mathbf{x}_0)^T \nabla^2 \Phi(\mathbf{x}_0)(\mathbf{x} - \mathbf{x}_0) \right\} \geq (>)$$
$$\text{Re} \left\{ [\nabla \Phi(\mathbf{x}_0) + (\mathbf{x} - \mathbf{x}_0)^T \nabla^2 \Phi(\mathbf{x}_0)] \Theta(\mathbf{x}, \mathbf{x}_0) \right\},$$

(ii) **(strictly)** $\Theta$**-pseudobonvex** *at* $\mathbf{x}_0 \in Q \subset \mathbb{C}^{2n}$ *if there exists a certain mapping* $\Theta : \mathbb{C}^{2n} \times \mathbb{C}^{2n} \to \mathbb{C}^{2n}$ *such that, for any* $\mathbf{x} \in Q$,

$$\text{Re} \left\{ [\nabla \Phi(\mathbf{x}_0) + (\mathbf{x} - \mathbf{x}_0)^T \nabla^2 \Phi(\mathbf{x}_0)] \Theta(\mathbf{x}, \mathbf{x}_0) \right\} \geq 0$$
$$\Rightarrow \text{Re} \left\{ \Phi(\mathbf{x}) - \Phi(\mathbf{x}_0) + \tfrac{1}{2}(\mathbf{x} - \mathbf{x}_0)^T \nabla^2 \Phi(\mathbf{x}_0)(\mathbf{x} - \mathbf{x}_0) \right\} \geq 0 \ (> 0),$$

(iii) $\Theta$**-quasibonvex** *at* $\mathbf{x}_0 \in Q$ *if there exists a certain mapping* $\Theta : \mathbb{C}^{2n} \times \mathbb{C}^{2n} \to \mathbb{C}^{2n}$ *such that, for any* $\mathbf{x} \in Q$,

$$\text{Re} \left\{ \Phi(\mathbf{x}) - \Phi(\mathbf{x}_0) + \tfrac{1}{2}(\mathbf{x} - \mathbf{x}_0)^T \nabla^2 \Phi(\mathbf{x}_0)(\mathbf{x} - \mathbf{x}_0) \right\} \leq 0$$
$$\Rightarrow \text{Re} \left\{ [\nabla \Phi(\mathbf{x}_0) + (\mathbf{x} - \mathbf{x}_0)^T \nabla^2 \Phi(\mathbf{x}_0)] \Theta(\mathbf{x}, \mathbf{x}_0) \right\} \leq 0.$$

## 3. Second-Order Mond–Weir Type Dual Model

We are going to establish two types of second-order parametric free dual model with respect to problem (P). These dual models are called the second-order Mond–Weir type dual model and the second-order Wolfe type dual model. For convenience, we give some symbols as follows. For $\mathbf{z} \in \mathbb{C}^{2n}$, $\mathbf{y}_i \in \mathbb{C}^{2m}$, $\mu \in \mathbb{C}^p$, the second-ordered differentiable functions $f$, $g : \mathbb{C}^{2n} \times \mathbb{C}^{2m} \to \mathbb{C}$ and $h : \mathbb{C}^{2n} \to \mathbb{C}$, we denote notations:

$$F^{(1)}(\mathbf{z}, \mathbf{y}_i) = \overline{\nabla_x f(\mathbf{z}, \mathbf{y}_i)} + \nabla_{\overline{x}} f(\mathbf{z}, \mathbf{y}_i);$$
$$F_1^{(2)}(\mathbf{z}, \mathbf{y}_i) = \overline{\nabla_{xx} f(\mathbf{z}, \mathbf{y}_i)} + \nabla_{\overline{xx}} f(\mathbf{z}, \mathbf{y}_i); \quad F_2^{(2)}(\mathbf{z}, \mathbf{y}_i) = \overline{\nabla_{\overline{x}x} f(\mathbf{z}, \mathbf{y}_i)} + \nabla_{x\overline{x}} f(\mathbf{z}, \mathbf{y}_i);$$
$$G^{(1)}(\mathbf{z}, \mathbf{y}_i) = \overline{\nabla_x g(\mathbf{z}, \mathbf{y}_i)} + \nabla_{\overline{x}} g(\mathbf{z}, \mathbf{y}_i);$$
$$G_1^{(2)}(\mathbf{z}, \mathbf{y}_i) = \overline{\nabla_{xx} g(\mathbf{z}, \mathbf{y}_i)} + \nabla_{\overline{xx}} g(\mathbf{z}, \mathbf{y}_i); \quad G_2^{(2)}(\mathbf{z}, \mathbf{y}_i) = \overline{\nabla_{\overline{x}x} g(\mathbf{z}, \mathbf{y}_i)} + \nabla_{x\overline{x}} g(\mathbf{z}, \mathbf{y}_i);$$
$$H^{(1)}(\mathbf{z}, \mu) = \mu^T \overline{\nabla_x h(\mathbf{z})} + \mu^H \nabla_{\overline{x}} h(\mathbf{z});$$
$$H_1^{(2)}(\mathbf{z}, \mu) = \mu^T \overline{\nabla_{xx} h(\mathbf{z})} + \mu^H \nabla_{\overline{xx}} h(\mathbf{z}); \quad H_2^{(2)}(\mathbf{z}, \mu) = \mu^T \overline{\nabla_{\overline{x}x} h(\mathbf{z})} + \mu^H \nabla_{x\overline{x}} h(\mathbf{z}).$$

The second-order Mond–Weir type dual problem (2nd-MWD) of problem (P) is a maximize problem as the following form:

$$(\text{2nd-MWD}) \quad \max_{(k,\widetilde{\lambda},\widetilde{\mathbf{y}})\in K(\mathbf{z})} \quad \max_{(\mathbf{z},\mu,\nu)\in X_1(k,\widetilde{\lambda},\widetilde{\mathbf{y}})} \quad \frac{\sum_{i=1}^{k} \lambda_i \operatorname{Re} f(\mathbf{z},\mathbf{y}_i)}{\sum_{i=1}^{k} \lambda_i \operatorname{Re} g(\mathbf{z},\mathbf{y}_i)},$$

where the set $K(\mathbf{z})$ is the collection of the component $(k,\widetilde{\lambda},\widetilde{\mathbf{y}})$ (here, $\widetilde{\lambda} = (\lambda_1,\ldots,\lambda_k)$, $\lambda_i \geq 0$ for $i = 1,\ldots,k$ with $\sum_{i=1}^{k} \lambda_i = 1$ and $\widetilde{\mathbf{y}} = (\mathbf{y}_1,\ldots,\mathbf{y}_k)$, $\mathbf{y}_i \in \mathbb{C}^{2m}$ for $i = 1,\ldots,k$) satisfying the necessary optimality conditions of problem (P) for any given feasible solution $\mathbf{z} = (z,\overline{z}) \in Q$ with constraint qualification holding, then there exists a nonzero multiplier $\mu \in T^* \subset \mathbb{C}^p$ such that $\operatorname{Re}\langle \mathbf{s},\mu\rangle \geq 0$ for $\mathbf{s} \in T$. Thus, $\operatorname{Re}\langle h(\mathbf{z}),\mu\rangle \leq 0$ as $-h(\mathbf{z}) \in T \subset \mathbb{C}^p$. The constraint set $X_1(k,\widetilde{\lambda},\widetilde{\mathbf{y}})$ is the collection of all feasible solutions $(\mathbf{z},\mu,\nu) \in \mathbb{C}^{2n} \times \mathbb{C}^p \times \mathbb{C}^n$ of (2nd-MWD), which satisfies the following expressions: for $\mathbf{z} = (z,\overline{z}) \in Q$ and $0 \neq \mu \in T^*$, such that

$$\begin{aligned}
&H^{(1)}(\mathbf{z},\mu) + \nu^H H_1^{(2)}(\mathbf{z},\mu) + \nu^T H_2^{(2)}(\mathbf{z},\mu) + \\
&\left\{ \sum_{i=1}^{k} \lambda_i \left[ F^{(1)}(\mathbf{z},\mathbf{y}_i) + \nu^H F_1^{(2)}(\mathbf{z},\mathbf{y}_i) + \nu^T F_2^{(2)}(\mathbf{z},\mathbf{y}_i) \right] \right\} \cdot \left( \sum_{i=1}^{k} \lambda_i \operatorname{Re} g(\mathbf{z},\mathbf{y}_i) \right) \\
&- \left( \sum_{i=1}^{k} \lambda_i \operatorname{Re} f(\mathbf{z},\mathbf{y}_i) \right) \cdot \left\{ \sum_{i=1}^{k} \lambda_i \left[ G^{(1)}(\mathbf{z},\mathbf{y}_i) + \nu^H G_1^{(2)}(\mathbf{z},\mathbf{y}_i) + \nu^T G_2^{(2)}(\mathbf{z},\mathbf{y}_i) \right] \right\} = 0,
\end{aligned} \tag{5}$$

$$\begin{aligned}
&\left( \sum_{i=1}^{k} \lambda_i \operatorname{Re} f(\mathbf{z},\mathbf{y}_i) \right) \cdot \left\{ \sum_{i=1}^{k} \lambda_i \operatorname{Re} \left\langle \nu, \nu^H G_1^{(2)}(\mathbf{z},\mathbf{y}_i) + \nu^T G_2^{(2)}(\mathbf{z},\mathbf{y}_i) \right\rangle \right\} \geq \\
&\left\{ \sum_{i=1}^{k} \lambda_i \operatorname{Re} \left\langle \nu, \nu^H F_1^{(2)}(\mathbf{z},\mathbf{y}_i) + \nu^T F_2^{(2)}(\mathbf{z},\mathbf{y}_i) \right\rangle \right\} \cdot \left( \sum_{i=1}^{k} \lambda_i \operatorname{Re} g(\mathbf{z},\mathbf{y}_i) \right),
\end{aligned} \tag{6}$$

$$\operatorname{Re}\langle h(\mathbf{z}),\mu\rangle \geq \tfrac{1}{2} \operatorname{Re} \left\langle \nu, \nu^H H_1^{(2)}(\mathbf{z},\mu) + \nu^T H_2^{(2)}(\mathbf{z},\mu) \right\rangle. \tag{7}$$

Denote a function

$$\Phi_1(\bullet) = \left[ \sum_{i=1}^{k} \lambda_i \operatorname{Re} f(\bullet,\mathbf{y}_i) \right] \cdot \left( \sum_{i=1}^{k} \lambda_i \operatorname{Re} g(\mathbf{z},\mathbf{y}_i) \right) - \left( \sum_{i=1}^{k} \lambda_i \operatorname{Re} f(\mathbf{z},\mathbf{y}_i) \right) \cdot \left[ \sum_{i=1}^{k} \lambda_i \operatorname{Re} g(\bullet,\mathbf{y}_i) \right].$$

The duality theorems of (2nd-MWD) with respect to primary problem (P) are established as follows. First, we will prove that the feasible value of (P) is not less than the feasible value of (2nd-MWD) under some suitable assumptions.

**Theorem 2** (Weak Duality). *Let* $\mathbf{x} = (x,\overline{x})$ *be a (P)-feasible solution,* $(k,\widetilde{\lambda},\widetilde{\mathbf{y}},\mathbf{z},\mu,\nu)$ *be (2nd-MWD)-feasible solution. If any one of the following conditions holds:*

(i)　$\Phi_1(\bullet)$ *is* $\Theta$-*pseudobonvex and* $\operatorname{Re}\langle h(\bullet),\mu\rangle$ *is* $\Theta$-*quasibonvex at* $\mathbf{z} \in Q$,
(ii)　$\Phi_1(\bullet)$ *is* $\Theta$-*quasibonvex and* $\operatorname{Re}\langle h(\bullet),\mu\rangle$ *is strictly* $\Theta$-*pseudobonvex at* $\mathbf{z} \in Q$,
(iii) $\Phi_1(\bullet)$ *and* $\operatorname{Re}\langle h(\bullet),\mu\rangle$ *are both* $\Theta$-*bonvex at* $\mathbf{z} \in Q$,

*then*

$$\sup_{\mathbf{y}\in Y} \frac{\operatorname{Re} f(\mathbf{x},\mathbf{y})}{\operatorname{Re} g(\mathbf{x},\mathbf{y})} \geq \frac{\sum_{i=1}^{k} \lambda_i \operatorname{Re} f(\mathbf{z},\mathbf{y}_i)}{\sum_{i=1}^{k} \lambda_i \operatorname{Re} g(\mathbf{z},\mathbf{y}_i)}.$$

**Proof.** Suppose, on the contrary, that

$$\sup_{\mathbf{y}\in Y} \frac{\operatorname{Re} f(\mathbf{x},\mathbf{y})}{\operatorname{Re} g(\mathbf{x},\mathbf{y})} < \frac{\sum_{i=1}^{k} \lambda_i \operatorname{Re} f(\mathbf{z},\mathbf{y}_i)}{\sum_{i=1}^{k} \lambda_i \operatorname{Re} g(\mathbf{z},\mathbf{y}_i)}.$$

Then, for all $\mathbf{y} \in Y$,

$$[\operatorname{Re} f(\mathbf{x}, \mathbf{y})] \cdot \Big\{ \sum_{i=1}^{k} \lambda_i \operatorname{Re} g(\mathbf{z}, \mathbf{y}_i) \Big\} < [\operatorname{Re} g(\mathbf{x}, \mathbf{y})] \cdot \Big\{ \sum_{i=1}^{k} \lambda_i \operatorname{Re} f(\mathbf{z}, \mathbf{y}_i) \Big\}.$$

Since $\lambda_i \geq 0$ with $\sum_{i=1}^{k} \lambda_1 = 1$ and given $\mathbf{y}_i \in Y$ for $i = 1, \cdots, k$, we obtain

$$[\sum_{i=1}^{k} \lambda_i \operatorname{Re} f(\mathbf{x}, \mathbf{y}_i)] \cdot \Big\{ \sum_{i=1}^{k} \lambda_i \operatorname{Re} g(\mathbf{z}, \mathbf{y}_i) \Big\} - [\sum_{i=1}^{k} \lambda_i \operatorname{Re} g(\mathbf{x}, \mathbf{y}_i)] \cdot \Big\{ \sum_{i=1}^{k} \lambda_i \operatorname{Re} f(\mathbf{x}, \mathbf{y}_i) \Big\} < 0.$$

The above inequality is $\Phi_1(\mathbf{x}) < 0$. Since $\Phi_1(\mathbf{z}) = 0$, we have

$$\Phi_1(\mathbf{x}) < \Phi_1(\mathbf{z}). \tag{8}$$

For $\mathbf{z} = (z, \bar{z})$, $\mathbf{x} = (x, \bar{x})$ and let $\mathbf{x} - \mathbf{z} = (v, \bar{v}) = (x - z, \overline{x - z})$, we know that Equation (6) is

$$(\mathbf{x} - \mathbf{z})^T \nabla^2 \Phi_1(\mathbf{z})(\mathbf{x} - \mathbf{z}) \leq 0. \tag{9}$$

On the other hand, if $\mathbf{x} = (x, \bar{x})$ is a feasible solution of problem (P), and since Theorem 1 holds, then there exists a nonzero multiplier $\mu \in T^* \subset \mathbb{C}^p$ such that $Re\langle \mathbf{s}, \mu \rangle \geq 0$ for $\mathbf{s} \in T$. Thus, the constraint condition of problem (P) could be expressed as

$$\operatorname{Re}\langle h(\mathbf{x}), \mu \rangle \leq 0.$$

If $(k, \widetilde{\lambda}, \widetilde{\mathbf{y}}, \mathbf{z}, \mu, v)$ is a feasible solution of (2nd-MWD), then Equation (7) holds. That is,

$$0 \leq \operatorname{Re}\langle h(\mathbf{z}), \mu \rangle - \frac{1}{2} \operatorname{Re} \Big\langle v, \, v^H H_1^{(2)}(\mathbf{z}, \mu) + v^T H_2^{(2)}(\mathbf{z}, \mu) \Big\rangle.$$

Therefore,

$$\operatorname{Re}\langle h(\mathbf{x}) - h(\mathbf{z}), \mu \rangle + \frac{1}{2} \operatorname{Re} \Big\langle v, \, v^H H_1^{(2)}(\mathbf{z}, \mu) + v^T H_2^{(2)}(\mathbf{z}, \mu) \Big\rangle \leq 0. \tag{10}$$

1.  From the hypotheses (*i*), $\Phi_1(\bullet)$ is $\Theta$-pseudobonvex at $\mathbf{z}$, and by Equations (8) and (9), there is a mapping $\Theta : \mathbb{C}^{2n} \times \mathbb{C}^{2n} \to \mathbb{C}^{2n}$ such that

$$\operatorname{Re} \Big\{ [\nabla \Phi_1(\mathbf{z}) + (\mathbf{x} - \mathbf{z})^T \nabla^2 \Phi_1(\mathbf{z})] \Theta(\mathbf{x}, \mathbf{z}) \Big\} < 0. \tag{11}$$

Since $\operatorname{Re}\langle h(\bullet), \mu \rangle$ is $\Theta$-quasibonvex at $\mathbf{z}$ and by Equation (10), we have

$$\operatorname{Re} \Big\{ [\nabla h(\mathbf{z}) + (\mathbf{x} - \mathbf{z})^T \nabla^2 h(\mathbf{z})] \Theta(\mathbf{x}, \mathbf{z}) \Big\} \leq 0. \tag{12}$$

By Equations (11) and (12), we obtain

$$\operatorname{Re} \Big\{ [(\nabla \Phi_1(\mathbf{z}) + \nabla h(\mathbf{z})) + (\mathbf{x} - \mathbf{z})^T (\nabla^2 \Phi_1(\mathbf{z}) + \nabla^2 h(\mathbf{z}))] \Theta(\mathbf{x}, \mathbf{z}) \Big\} < 0.$$

Thus,

$$\begin{aligned} &H^{(1)}(\mathbf{z}, \mu) + v^H H_1^{(2)}(\mathbf{z}, \mu) + v^T H_2^{(2)}(\mathbf{z}, \mu) + \\ &\Big\{ \sum_{i=1}^{k} \lambda_i [F^{(1)}(\mathbf{z}, \mathbf{y}_i) + v^H F_1^{(2)}(\mathbf{z}, \mathbf{y}_i) + v^T F_2^{(2)}(\mathbf{z}, \mathbf{y}_i)] \Big\} \cdot (\sum_{i=1}^{k} \lambda_i \operatorname{Re} g(\mathbf{z}, \mathbf{y}_i)) - \\ &(\sum_{i=1}^{k} \lambda_i \operatorname{Re} f(\mathbf{z}, \mathbf{y}_i)) \cdot \Big\{ \sum_{i=1}^{k} \lambda_i [G^{(1)}(\mathbf{z}, \mathbf{y}_i) + v^H G_1^{(2)}(\mathbf{z}, \mathbf{y}_i) + v^T G_2^{(2)}(\mathbf{z}, \mathbf{y}_i)] \Big\} \neq 0. \end{aligned} \tag{13}$$

This contradicts the equality of Equation (5).

2.  If hypothesis (*ii*) is true, $\Phi_1(\bullet)$ is $\Theta$-quasibonvex at $\mathbf{z}$, then Equation (11) becomes less than or equal to zero. Since Re $\langle h(\bullet), \mu \rangle$ is strictly $\Theta$-pseudobonvex at $\mathbf{z}$, then Equation (12) becomes less than zero. Thus, Equation (13) holds, and it still contradicts the equality of Equation (5).

3.  Suppose that the hypothesis (*iii*) is true. By Equations (8) and (9), we have

$$\text{Re}\left\{ [\Phi_1(\mathbf{x}) - \Phi_1(\mathbf{z})] + \frac{1}{2}(\mathbf{x} - \mathbf{z})^T \nabla^2 \Phi_1(\mathbf{z})(\mathbf{x} - \mathbf{z}) \right\} < 0.$$

If $\Phi_1(\bullet)$ is $\Theta$-bonvex at $\mathbf{z} \in Q$, and from the above inequality, then there is a mapping $\Theta : \mathbb{C}^{2n} \times \mathbb{C}^{2n} \to \mathbb{C}^{2n}$ such that the Equation (11) holds. From Equation (10) and if Re $\langle h(\bullet), \mu \rangle$ is $\Theta$-bonvex at $\mathbf{z} \in Q$, then we obtain Equation (12). Equation (13) still holds, and it contradicts the equality of Equation (5).

Therefore, the result of the theorem is proved.　□

Given an optimal solution of problem (P), we can obtain a feasible solution of the dual problem (2nd-MWD), and the following strong duality theorem will be proved.

**Theorem 3** (Strong Duality). *Let $\mathbf{x}_0 = (x_0, \overline{x_0})$ be an optimal solution of problem (P). Then, there are $(k, \widetilde{\lambda}, \widetilde{\mathbf{y}}) \in K(\mathbf{x}_0)$ and $(\mathbf{x}_0, \mu, v) \in X(k, \widetilde{\lambda}, \widetilde{\mathbf{y}})$ such that $(k, \widetilde{\lambda}, \widetilde{\mathbf{y}}, \mathbf{x}_0, \mu, v)$ is a feasible of the dual problem (2nd-MWD). If the hypotheses of a weak duality theorem are fulfilled, then $(k, \widetilde{\lambda}, \widetilde{\mathbf{y}}, \mathbf{x}_0, \mu, v)$ is an optimal solution of (2nd-MWD), and problems (P) and (2nd-MWD) have the same optimal values.*

**Proof.** Let $\mathbf{x}_0$ be an optimal solution of (P) with optimal value

$$\gamma^* = \Psi(\mathbf{x}_0) = \frac{\sum_{i=1}^{k} \lambda_i \text{Re} f(\mathbf{x}_0, \mathbf{y}_i)}{\sum_{i=1}^{k} \lambda_i \text{Re} g(\mathbf{x}_0, \mathbf{y}_i)}.$$

From Theorem 1, we could obtain the nonzero $\mu \in T^* \subset \mathbb{C}^p$, positive integer $k$ with $\mathbf{y}_i \in Y(\mathbf{x}_0)$, multipliers $\lambda_i \geq 0$ for $i = 1, \dots, k$ and $\sum_{i=1}^{k} \lambda_i = 1$ such that

$$\left\{ \sum_{i=1}^{k} \lambda_i \left[ \overline{\nabla_x f(\mathbf{x}_0, \mathbf{y}_i)} + \nabla_{\overline{x}} f(\mathbf{x}_0, \mathbf{y}_i) \right] + \mu^T \ \overline{\nabla_x h(\mathbf{x}_0)} + \mu^H \nabla_{\overline{x}} h(\mathbf{x}_0) \right\} \times \left( \sum_{i=1}^{k} \lambda_i \ \text{Re} \ g(\mathbf{x}_0, \mathbf{y}_i) \right)$$

$$- \left( \sum_{i=1}^{k} \lambda_i \ \text{Re} \ f(\mathbf{x}_0, \mathbf{y}_i) \right) \times \left\{ \sum_{i=1}^{k} \lambda_i \left[ \overline{\nabla_x g(\mathbf{x}_0, \mathbf{y}_i)} + \nabla_{\overline{x}} g(\mathbf{x}_0, \mathbf{y}_i) \right] \right\} = 0.$$

If we take $v = x_0 - x_0 = 0$ and replace $\mu$ by $\mu \times \left( \sum_{i=1}^{k} \lambda_i \ \text{Re} \ g(\mathbf{x}_0, \mathbf{y}_i) \right)$, then

$$H^{(1)}(\mathbf{x}_0, \mu) + v^H H_1^{(2)}(\mathbf{x}_0, \mu) + v^T H_2^{(2)}(\mathbf{x}_0, \mu) +$$

$$\left\{ \sum_{i=1}^{k} \lambda_i \left[ F^{(1)}(\mathbf{x}_0, \mathbf{y}_i) + v^H F_1^{(2)}(\mathbf{x}_0, \mathbf{y}_i) + v^T F_2^{(2)}(\mathbf{x}_0, \mathbf{y}_i) \right] \right\} \cdot \left( \sum_{i=1}^{k} \lambda_i \text{Re} \ g(\mathbf{x}_0, \mathbf{y}_i) \right)$$

$$- \left( \sum_{i=1}^{k} \lambda_i \text{Re} \ f(\mathbf{x}_0, \mathbf{y}_i) \right) \cdot \left\{ \sum_{i=1}^{k} \lambda_i \left[ G^{(1)}(\mathbf{x}_0, \mathbf{y}_i) + v^H G_1^{(2)}(\mathbf{x}_0, \mathbf{y}_i) + v^T G_2^{(2)}(\mathbf{x}_0, \mathbf{y}_i) \right] \right\} = 0,$$

and the component $(\mathbf{x}_0, \mu, v = 0) \in X_1(k, \widetilde{\lambda}, \widetilde{\mathbf{y}})$ is satisfying conditions Equations (5)–(7) of problem (2nd-MWD). It follows that $(k, \widetilde{\lambda}, \widetilde{\mathbf{y}}, \mathbf{x}_0, \mu, v = 0)$ is a feasible solution of (2nd-MWD). If the hypotheses of Theorem 2 are fulfilled, then $(k, \widetilde{\lambda}, \widetilde{\mathbf{y}}, \mathbf{x}_0, \mu, v = 0)$ is an optimal solution of (2nd-MWD), and the two problems (P) and (2nd-MWD) have the same optimal values.　□

If both optimal solutions of primary problem (P) and dual problem (2nd-MWD) exist, then the optimal values of (P) and (2nd-MWD) are equal under some assumptions. We could prove this result as the following theorem.

**Theorem 4.** *(Strictly Converse Duality) Let* $\mathbf{x}$ *and* $(k, \widetilde{\lambda}, \widetilde{\mathbf{y}}, \mathbf{z}, \mu, v)$ *be optimal solutions of (P) and (2nd-MWD), and assume that the assumptions of strong duality theorem are fulfilled. In addition, if* $\Phi_1(\bullet)$ *is strictly* $\Theta$*-pseudobonvex and* $\mathrm{Re}\langle h(\bullet), \mu \rangle$ *is* $\Theta$*-quasibonvex at* $\mathbf{z} \in Q$*, then* $\mathbf{x} = \mathbf{z}$*; and the optimal values of (P) and (2nd-MWD) are equal.*

**Proof.** Assume that $\mathbf{x} \neq \mathbf{z}$, and reach a contradiction.

By strong duality theorem (Theorem 3),

$$\sup_{\mathbf{y} \in Y} \frac{\mathrm{Re}\, f(\mathbf{x}, \mathbf{y})}{\mathrm{Re}\, g(\mathbf{x}, \mathbf{y})} = \frac{\sum_{i=1}^{k} \lambda_i \, \mathrm{Re}\, f(\mathbf{z}, \mathbf{y}_i)}{\sum_{i=1}^{k} \lambda_i \, \mathrm{Re}\, g(\mathbf{z}, \mathbf{y}_i)}.$$

Then, for all $\mathbf{y} \in Y$,

$$[\mathrm{Re}\, f(\mathbf{x}, \mathbf{y})] \cdot \Big\{ \sum_{i=1}^{k} \lambda_i \, \mathrm{Re}\, g(\mathbf{z}, \mathbf{y}_i) \Big\} \leq [\mathrm{Re}\, g(\mathbf{x}, \mathbf{y})] \cdot \Big\{ \sum_{i=1}^{k} \lambda_i \, \mathrm{Re}\, f(\mathbf{z}, \mathbf{y}_i) \Big\}.$$

Since $\lambda_i \geq 0$ with $\sum_{i=1}^{k} \lambda_1 = 1$ and given $\mathbf{y}_i \in Y$ for $i = 1, \cdots, k$, we have that

$$\Big[ \sum_{i=1}^{k} \lambda_i \mathrm{Re}\, f(\mathbf{x}, \mathbf{y}_i) \Big] \cdot \Big\{ \sum_{i=1}^{k} \lambda_i \, \mathrm{Re}\, g(\mathbf{z}, \mathbf{y}_i) \Big\} - \Big[ \sum_{i=1}^{k} \lambda_i \mathrm{Re}\, g(\mathbf{x}, \mathbf{y}_i) \Big] \cdot \Big\{ \sum_{i=1}^{k} \lambda_i \, \mathrm{Re}\, f(\mathbf{z}, \mathbf{y}_i) \Big\} \leq 0.$$

That is,

$$\Phi_1(\mathbf{x}) - \Phi_1(\mathbf{z}) \leq 0.$$

If $\Phi_1(\bullet)$ is strictly $\Theta$-pseudobonvex at $\mathbf{z}$, then, by using a similar process of the proof as in Theorem 2, we get Equation (11). Since $\mathrm{Re}\langle h(\bullet), \mu \rangle$ is $\Theta$-quasibonvex at $\mathbf{z}$, we obtain Equation (12). By Equations (11) and (12), Equation (13) still holds. It contradicts the equality of Equation (5). This is the complete proof.　□

## 4. Second-Order Wolfe Type Dual Model

The second-order Wolfe type dual problem with respect to problem (P) is the following form:

$$(\text{2nd-WD}) \quad \max_{(k, \widetilde{\lambda}, \widetilde{\mathbf{y}}) \in K(\mathbf{z})} \quad \max_{(\mathbf{z}, \mu, v) \in X_2(k, \widetilde{\lambda}, \widetilde{\mathbf{y}})} \quad \frac{\displaystyle\sum_{i=1}^{k} \lambda_i \, \mathrm{Re}\, [f(\mathbf{z}, \mathbf{y}_i) + \langle h(\mathbf{z}), \mu \rangle]}{\displaystyle\sum_{i=1}^{k} \lambda_i \, \mathrm{Re}\, g(\mathbf{z}, \mathbf{y}_i)},$$

where the set $K(\mathbf{z})$ is the collection of the component $(k, \widetilde{\lambda}, \widetilde{\mathbf{y}})$ (here $\widetilde{\lambda} = (\lambda_1, \ldots, \lambda_k)$, $\lambda_i \geq 0$ for $i = 1, \ldots, k$ with $\sum_{i=1}^{k}$ and $\widetilde{\mathbf{y}} = (\mathbf{y}_1, \ldots, \mathbf{y}_k)$, $\mathbf{y}_i \in \mathbb{C}^{2m}$ for $i = 1, \ldots, k$) satisfying the necessary optimality conditions of problem (P) for any given feasible solution $\mathbf{z} = (z, \bar{z}) \in Q$ with constraint qualification hold, then there exists a nonzero multiplier $\mu \in T^* \subset \mathbb{C}^p$ such that $Re\langle \mathbf{s}, \mu \rangle \geq 0$ for $\mathbf{s} \in T$. Thus, $\mathrm{Re}\langle h(\mathbf{z}), \mu \rangle \leq 0$ as $-h(\mathbf{z}) \in T \subset \mathbb{C}^p$. The constraint set $X_2(k, \widetilde{\lambda}, \widetilde{\mathbf{y}})$ satisfies the following conditions: for $\mathbf{z} = (z, \bar{z}) \in Q$ and $0 \neq \mu \in T^*$, such that

$$\begin{aligned} &\Big\{ \textstyle\sum_{i=1}^{k} \lambda_i \big( [F^{(1)}(\mathbf{z}, \mathbf{y}_i) + H^{(1)}(\mathbf{z}, \mu)] + \\ &v^H [F_1^{(2)}(\mathbf{z}, \mathbf{y}_i) + H_1^{(2)}(\mathbf{z}, \mu)] + v^T [F_2^{(2)}(\mathbf{z}, \mathbf{y}_i) + H_2^{(2)}(\mathbf{z}, \mu)] \big) \Big\} \times \big( \textstyle\sum_{i=1}^{k} \lambda_i \mathrm{Re}\, g(\mathbf{z}, \mathbf{y}_i) \big) \\ &- \big( \textstyle\sum_{i=1}^{k} \lambda_i \mathrm{Re}[f(\mathbf{z}, \mathbf{y}_i) + \langle h(\mathbf{z}), \mu \rangle] \big) \times \big\{ \textstyle\sum_{i=1}^{k} \lambda_i [G^{(1)}(\mathbf{z}, \mathbf{y}_i) + v^H G_1^{(2)}(\mathbf{z}, \mathbf{y}_i) + v^T G_2^{(2)}(\mathbf{z}, \mathbf{y}_i)] \big\} = 0, \end{aligned} \tag{14}$$

$$\begin{aligned} &\big( \textstyle\sum_{i=1}^{k} \lambda_i Re[f(\mathbf{z}, \mathbf{y}_i) + \langle h(\mathbf{z}), \mu \rangle] \big) \times \big\{ \textstyle\sum_{i=1}^{k} \lambda_i \mathrm{Re}\, \langle v, v^H G_1^{(2)}(\mathbf{z}, \mathbf{y}_i) + v^T G_2^{(2)}(\mathbf{z}, \mathbf{y}_i) \rangle \big\} \geq \\ &\big\{ \textstyle\sum_{i=1}^{k} \lambda_i \mathrm{Re}\, \langle v, v^H [F_1^{(2)}(\mathbf{z}, \mathbf{y}_i) + H_1^{(2)}(\mathbf{z}, \mu)] + v^T [F_2^{(2)}(\mathbf{z}, \mathbf{y}_i) + H_2^{(2)}(\mathbf{z}, \mu)] \rangle \big\} \times \big( \textstyle\sum_{i=1}^{k} \lambda_i \mathrm{Re}\, g(\mathbf{z}, \mathbf{y}_i) \big). \end{aligned} \tag{15}$$

Denote function $\Phi_2(\bullet)$ by

$$\Phi_2(\bullet) = \left\{ \sum_{i=1}^{k} \lambda_i \mathrm{Re}\, [f(\bullet, \mathbf{y}_i) + \langle h(\bullet), \mu \rangle] \right\} \times \left( \sum_{i=1}^{k} \lambda_i \mathrm{Re}\, g(\mathbf{z}, \mathbf{y}_i) \right)$$
$$- \left( \sum_{i=1}^{k} \lambda_i \mathrm{Re}\, [f(\mathbf{z}, \mathbf{y}_i) + \langle h(\mathbf{z}), \mu \rangle] \right) \times \left\{ \sum_{i=1}^{k} \lambda_i \mathrm{Re}\, g(\bullet, \mathbf{y}_i) \right\}.$$

We could state and prove the duality theorems of (2nd-WD) under the second-order generalized $\Theta$-bonvexities as follows.

**Theorem 5** (Weak Duality). *Let* $\mathbf{x} = (x, \bar{x})$ *be (P)-feasible solution,* $(k, \tilde{\lambda}, \tilde{\mathbf{y}}, \mathbf{z}, \mu, \nu)$ *be (2nd-WD)-feasible solution, and if* $\Phi_2(\bullet)$ *is* $\Theta$*-pseudobonvex at* $\mathbf{z} \in Q$. *Then,*

$$\max_{\mathbf{y} \in Y} \frac{\mathrm{Re}\, f(\mathbf{x}, \mathbf{y})}{\mathrm{Re}\, g(\mathbf{x}, \mathbf{y})} \geq \frac{\sum_{i=1}^{k} \lambda_i \,\mathrm{Re}\, [f(\mathbf{z}, \mathbf{y}_i) + \langle h(\mathbf{z}), \mu \rangle]}{\sum_{i=1}^{k} \lambda_i \,\mathrm{Re}\, g(\mathbf{z}, \mathbf{y}_i)}.$$

**Proof.** Suppose, on the contrary, that

$$\max_{\mathbf{y} \in Y} \frac{\mathrm{Re}\, f(\mathbf{x}, \mathbf{y})}{\mathrm{Re}\, g(\mathbf{x}, \mathbf{y})} < \frac{\sum_{i=1}^{k} \lambda_i \,\mathrm{Re}\, [f(\mathbf{z}, \mathbf{y}_i) + \langle h(\mathbf{z}), \mu \rangle]}{\sum_{i=1}^{k} \lambda_i \,\mathrm{Re}\, g(\mathbf{z}, \mathbf{y}_i)}.$$

Then, for all $\mathbf{y} \in Y$,

$$[\mathrm{Re}\, f(\mathbf{x}, \mathbf{y})] \cdot \left\{ \sum_{i=1}^{k} \lambda_i \,\mathrm{Re}\, g(\mathbf{z}, \mathbf{y}_i) \right\} < [\mathrm{Re}\, g(\mathbf{x}, \mathbf{y})] \cdot \left\{ \sum_{i=1}^{k} \lambda_i \,\mathrm{Re}\, [f(\mathbf{z}, \mathbf{y}_i) + \langle h(\mathbf{z}), \mu \rangle] \right\}.$$

Since $\lambda_i \geq 0$ with $\sum_{i=1}^{k} \lambda_1 = 1$ and given $\mathbf{y}_i \in Y$ for $i = 1, \cdots, k$, we have that

$$[\sum_{i=1}^{k} \lambda_i \mathrm{Re}\, f(\mathbf{x}, \mathbf{y}_i)] \cdot \left\{ \sum_{i=1}^{k} \lambda_i \,\mathrm{Re}\, g(\mathbf{z}, \mathbf{y}_i) \right\} - [\sum_{i=1}^{k} \lambda_i \mathrm{Re}\, g(\mathbf{x}, \mathbf{y}_i)] \cdot \left\{ \sum_{i=1}^{k} \lambda_i \,\mathrm{Re}\, [f(\mathbf{z}, \mathbf{y}_i) + \langle h(\mathbf{z}), \mu \rangle] \right\} < 0.$$

Let $\mathbf{x} = (x, \bar{x})$ be the feasible solution of (P) that is

$$\mathrm{Re}\langle h(\mathbf{x}), \mu \rangle \leq 0.$$

By the above two inequalities, we get

$$\left\{ \sum_{i=1}^{k} \lambda_i \mathrm{Re}\, [f(\mathbf{x}, \mathbf{y}_i) + \langle h(\mathbf{x}), \mu \rangle] \right\} \cdot \left\{ \sum_{i=1}^{k} \lambda_i \,\mathrm{Re}\, g(\mathbf{z}, \mathbf{y}_i) \right\}$$
$$- [\sum_{i=1}^{k} \lambda_i \mathrm{Re}\, g(\mathbf{x}, \mathbf{y}_i)] \cdot \left\{ \sum_{i=1}^{k} \lambda_i \,\mathrm{Re}\, [f(\mathbf{z}, \mathbf{y}_i) + \langle h(\mathbf{z}), \mu \rangle] \right\} < 0.$$

This implies that

$$\Phi_2(\mathbf{x}) < 0 = \Phi_2(\mathbf{z}). \tag{16}$$

For $\mathbf{z} = (z, \bar{z}), \mathbf{x} = (x, \bar{x})$ and let $\mathbf{x} - \mathbf{z} = (v, \bar{v}) = (x - z, \overline{x - z})$. From Lemma 2,

$$\mathrm{Re}\, [(\mathbf{x} - \mathbf{z})^T \nabla^2 \Phi_2(\mathbf{z})(\mathbf{x} - \mathbf{z})] = \left\{ \sum_{i=1}^{k} \lambda_i \mathrm{Re}\, \left\langle \nu,\, \nu^H [F_1^{(2)}(\mathbf{z}, \mathbf{y}_i) + H_1^{(2)}(\mathbf{z}, \mu)] \right. \right.$$
$$\left. + \nu^T [F_2^{(2)}(\mathbf{z}, \mathbf{y}_i) + H_2^{(2)}(\mathbf{z}, \mu)] \right\rangle \right\} \times \left( \sum_{i=1}^{k} \lambda_i \mathrm{Re}\, g(\mathbf{z}, \mathbf{y}_i) \right)$$
$$- \left( \sum_{i=1}^{k} \lambda_i Re[f(\mathbf{z}, \mathbf{y}_i) + \langle h(\mathbf{z}), \mu \rangle] \right) \times \left\{ \sum_{i=1}^{k} \lambda_i \mathrm{Re}\, \left\langle \nu,\, \nu^H G_1^{(2)}(\mathbf{z}, \mathbf{y}_i) + \nu^T G_2^{(2)}(\mathbf{z}, \mathbf{y}_i) \right\rangle \right\}.$$

From condition Equations (15) and (16) and the above inequality, we know that

$$\text{Re}\Big\{\Phi_2(\mathbf{x}) - \Phi_2(\mathbf{z}) + \frac{1}{2}(\mathbf{x}-\mathbf{z})^T\nabla^2\Phi_2(\mathbf{z})(\mathbf{x}-\mathbf{z})\Big\} < 0.$$

If $\Phi_2(\bullet)$ is $\Theta$-pseudobonvex at $\mathbf{z}$, then there is a mapping $\Theta : \mathbb{C}^{2n} \times \mathbb{C}^{2n} \to \mathbb{C}^{2n}$ such that

$$\text{Re}\Big\{[\nabla\Phi_2(\mathbf{z}) + (\mathbf{x}-\mathbf{z})^T\nabla^2\Phi_2(\mathbf{z})]\Theta(\mathbf{x},\mathbf{z})\Big\} < 0.$$

Thus,

$$
\begin{aligned}
&\Big\{ \textstyle\sum_{i=1}^{k}\lambda_i\big([F^{(1)}(\mathbf{z},\mathbf{y}_i) + H^{(1)}(\mathbf{z},\mu)] + v^H[F_1^{(2)}(\mathbf{z},\mathbf{y}_i) + H_1^{(2)}(\mathbf{z},\mu)] \\
&+ v^T[F_2^{(2)}(\mathbf{z},\mathbf{y}_i) + H_2^{(2)}(\mathbf{z},\mu)]\big)\Big\} \times \big(\textstyle\sum_{i=1}^{k}\lambda_i\text{Re }g(\mathbf{z},\mathbf{y}_i)\big) \\
&- \big(\textstyle\sum_{i=1}^{k}\lambda_i Re[f(\mathbf{z},\mathbf{y}_i) + \langle h(\mathbf{z}),\mu\rangle]\big) \times \Big\{\textstyle\sum_{i=1}^{k}\lambda_i\big[G^{(1)}(\mathbf{z},\mathbf{y}_i) + v^H G_1^{(2)}(\mathbf{z},\mathbf{y}_i) + v^T G_2^{(2)}(\mathbf{z},\mathbf{y}_i)\big]\Big\} \neq 0.
\end{aligned}
\tag{17}
$$

This contradicts the condition of Equation (14) in dual problem (2nd-WD). This is the complete proof. □

**Theorem 6** (Strong Duality). *Let* $\mathbf{x}_0 = (x_0, \overline{x_0})$ *be an optimal solution of problem (P). Then, there are* $(k, \widetilde{\lambda}, \widetilde{\mathbf{y}}) \in K(\mathbf{x}_0)$ *and* $(\mathbf{x}_0, \mu, v) \in X(k, \widetilde{\lambda}, \widetilde{\mathbf{y}})$ *such that* $(k, \widetilde{\lambda}, \widetilde{\mathbf{y}}, \mathbf{x}_0, \mu, v)$ *is a feasible solution of the dual problem (2nd-WD). If the hypotheses of weak duality theorem are fulfilled, then* $(k, \widetilde{\lambda}, \widetilde{\mathbf{y}}, \mathbf{x}_0, \mu, v)$ *is an optimal solution of (2nd-WD), and problems (P) and (2nd-WD) have the same optimal values.*

**Proof.** It follows by the same way as the proof of strong duality theorem in (2nd-MWD). □

**Theorem 7** (Strictly Converse Duality). *Let* $\mathbf{x}$ *and* $(k, \widetilde{\lambda}, \widetilde{\mathbf{y}}, \mathbf{z}, \mu, v)$ *be optimal solutions of (P) and (2nd-WD), and assume that the assumptions of a strong duality theorem are fulfilled. In addition, if* $\Phi_2(\bullet)$ *is strictly* $\Theta$-*pseudobonvex at* $\mathbf{z} \in Q$, *then* $\mathbf{x} = \mathbf{z}$, *and the optimal values of (P) and (2nd-WD) are equal.*

**Proof.** Assume that $\mathbf{x} \neq \mathbf{z}$ and reach a contradiction.
By strong duality theorem (Theorem 6),

$$\sup_{\mathbf{y}\in Y} \frac{\text{Re } f(\mathbf{x},\mathbf{y})}{\text{Re } g(\mathbf{x},\mathbf{y})} = \frac{\sum_{i=1}^{k}\lambda_i\text{Re }[f(\mathbf{z},\mathbf{y}_i) + \langle h(\mathbf{z}),\mu\rangle]}{\sum_{i=1}^{k}\lambda_i\text{Re }g(\mathbf{z},\mathbf{y}_i)}.$$

Then, for all $\mathbf{y} \in Y$,

$$[\text{Re } f(\mathbf{x},\mathbf{y})] \cdot \Big\{\sum_{i=1}^{k}\lambda_i\text{Re }g(\mathbf{z},\mathbf{y}_i)\Big\} \le [\text{Re }g(\mathbf{x},\mathbf{y})] \cdot \Big\{\sum_{i=1}^{k}\lambda_i\text{Re }[f(\mathbf{z},\mathbf{y}_i) + \langle h(\mathbf{z}),\mu\rangle]\Big\}.$$

Since $\lambda_i \ge 0$ with $\sum_{i=1}^{k}\lambda_1 = 1$ and given $\mathbf{y}_i \in Y$ for $i = 1,\dots,k$, we have that

$$[\sum_{i=1}^{k}\lambda_i\text{Re }f(\mathbf{x},\mathbf{y}_i)] \cdot \Big\{\sum_{i=1}^{k}\lambda_i\text{Re }g(\mathbf{z},\mathbf{y}_i)\Big\} - [\sum_{i=1}^{k}\lambda_i\text{Re }g(\mathbf{x},\mathbf{y})] \cdot \Big\{\sum_{i=1}^{k}\lambda_i\text{Re }[f(\mathbf{z},\mathbf{y}_i) + \langle h(\mathbf{z}),\mu\rangle]\Big\} \le 0.$$

By a similar process as the proof in Theorem 5, we can obtain

$$\Phi_2(\mathbf{x}) - \Phi_2(\mathbf{z}) \le 0$$

and

$$Re\Big\{\Phi_2(\mathbf{x}) - \Phi_2(\mathbf{z}) + \frac{1}{2}(\mathbf{x}-\mathbf{z})^T\nabla^2\Phi_2(\mathbf{z})(\mathbf{x}-\mathbf{z})\Big\} \le 0.$$

If $\Phi_2(\bullet)$ is strictly $\Theta$-pseudobonvex at **z**, then we obtain Equation (17); this contradicts the condition of Equation (14) in dual problem (2nd-WD). Therefore, the result of the theorem is proved. $\square$

## 5. Conclusions and Further Plausible Work

In this paper, we formulated the second-order Mond–Weir type and Wolfe type dual models with respect to problem (P), and derived their duality theorems. In further plausible work, we will establish the second-order mixed type dual problem (2nd-MD) of problem (P), and then we would like to show that the dual problems (2nd-MWD) and (2nd-WD) are the special cases of dual problem (2nd-MD).

**Funding:** This research was supported by MOST 108-2115-M-035-005-, Taiwan.

**Acknowledgments:** The authors express their sincere gratitude to the unknown reviewers for their detailed reading and valuable advice.

**Conflicts of Interest:** The author declares no conflict of interest.

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
