# Peer review of "Second-Order Parametric Free Dualities for Complex Minimax Fractional Programming"

_mathematics, doi:10.3390/math8010067_

Round 1

Reviewer 1 Report

In the current version of the paper most of the raised issues have been fixed. The language has been improved, but there are still some small mistakes, e.g.,
* line 29, in formal language one does not use "doesn't", it should be "does not",
* line 35, at the beginning there should be "has" instead of "have",
* line 55, it should be "from [7,12]" and not "in [7,12]",
* line 117, it should be "we give" and not "we given",
* line 175, it should be "Let x_0 be" and not "Let x_0 is",
* line 179, when giving a range one should use - and not the approx symbol, so it should be "(5)-(7)",
* line 183, at the end we find "are exist", but it should be without the "are", i.e., "exist",
* line 187 and 232, it should be "be optimal" and not "are optimal",
* line 195, we find "then use similar...", but it is better to write "then by using a similar...".

The proof of Lemma 3 has not been corrected. Still, at the beginning the author assumes some inequality that one cannot infer from the assumptions made in the lemma (the only assumption in this lemma is that Re[Phi] is Theta-bonvex). It should be clear from which facts this inequality was inferred. Without any justification one needs also to consider the other inequalities (>= in 1, and > in 2) to complete the proof.

Author Response

Response to Reviewer 1 Comments

Point 1: In the current version of the paper most of the raised issues have been fixed. The language has been improved, but there are still some small mistakes, e.g.,

* line 29, in formal language one does not use "doesn't", it should be "does not",

* line 35, at the beginning there should be "has" instead of "have",

* line 55, it should be "from [7,12]" and not "in [7,12]",

* line 117, it should be "we give" and not "we given",

* line 175, it should be "Let x_0 be" and not "Let x_0 is",

* line 179, when giving a range one should use - and not the approx symbol, so it should be "(5)-(7)",

* line 183, at the end we find "are exist", but it should be without the "are", i.e., "exist",

* line 187 and 232, it should be "be optimal" and not "are optimal",

* line 195, we find "then use similar...", but it is better to write "then by using a similar...". 

 Response 1:  We have revised English language mistakes. Please see lines: 29, 35, 55, 100, 156, 160, 164, 168, 213 and 176.

Point 2: The proof of Lemma 3 has not been corrected. Still, at the beginning the author assumes some inequality that one cannot infer from the assumptions made in the lemma (the only assumption in this lemma is that Re[Phi] is Theta-bonvex). It should be clear from which facts this inequality was inferred. Without any justification one needs also to consider the other inequalities (>= in 1, and > in 2) to complete the proof.

 Response 2: Because we used the Lemma 3 only in Theorem 2(iii). For the correctness of this paper, we are deleted the Lemma 3, and we re-proof the Theorem 2(iii) without use Lemma 3, please see lines: 143-147.

My revision paper is in Report Notes.

Reviewer 2 Report

The references [16] and [19] (but also others) must be corrected (the numbering of the pages is absent). Also, reference [20] must be completed with volume and page number.

The present manuscript can be accepted for publication in the journal after aforementioned remarks are considered.

Author Response

Response to Reviewer 2 Comments

Point 1: The references [16] and [19] (but also others) must be corrected (the numbering of the pages is absent). Also, reference [20] must be completed with volume and page number. 

Response 1: We have checked and improved all references.

My references of the revision paper is in Report Notes.

Reviewer 3 Report

The author established the 3 second-order Mond-Weir type and Wolfe type dual models and provided some minimax theorems in complex spaces.

It makes a very good impression, the detailed literature review on this current topic.

The Theorem 2. (Weak Duality) , Theorem 3. (Strong Duality) and Theorem 4. (Strictly Converse Duality) for the Second-order Mond-Weir type dual model and Theorems 5-7 for the Second-order Wolfe type dual model are proven to be precise and very professional.

Obviously, the research are of interest to specialists which are working in this professional field.

Obviously, the results obtained in this article can be continued successfully for establishing of the second-order mixed type dual problems.

This give me a reason to recommend the paper “Second-order parametric free dualities for complex minimax fractional programming” of Tone-Yau Huang for publishing in authoritative journal “Mathematics”.

Author Response

Thank you very much for your suggestion.

Reviewer 4 Report

The present paper investigates a minimax fractional programming in complex spaces. More precisely, by using the duality theory, the author establish weak, strong and strictly converse duality results associated with second-order Mond-Weir type and Wolfe type dual models.

The results seem correct and new and the topic described in the paper seems to be interesting. Also, the paper is well motivated and structured. It is self-contained and ordered.

In conclusion, I recommend publication in Mathematics.

Author Response

Thank you very much for your suggestion.

This manuscript is a resubmission of an earlier submission. The following is a list of the peer review reports and author responses from that submission.

Round 1

Reviewer 1 Report

Review report on “Second-order parametric free dualities for complex

minimax fractional programming” by Tone-Yau Huang

In the present paper, weak, strong and strictly converse duality theorems are established for a minimax fractional programming problem in complex spaces.

However, more relevant works should be references and some language issues should be clarified. I will recommend this paper for publication only if all the following comments and suggestions are addressed satisfactorily.

The introduction is without sufficient background on the relevant research subject. More backgrounds and references on the optimization, minimax theory and, in particular, their applications should be added. In this regard (fractional variational problems), the author is recommended the following references: ‘Efficiency conditions in vector control problems governed by multiple integrals’, ‘Duality with (\rho, b)-quasiinvexity for multidimensional vector fractional control problems’, ‘Multiobjective fractional variational problem on higher-order jet bundles’ and ‘Duality results in quasiinvex variational control problems with curvilinear integral functionals’. No space at the beginning of the following lines: 15, 21, 33, 52, 56, 59, 70, 71, 117, 172, 195. Line 2: "Duality model in programming problem plays ..." should be "Since duality model in programming problem plays ..." Line 7: "… have ..." should be "… are ..." Line 7: “… game theory and minimax programming …" should be “… game theory, minimax programming …" Line 9: “Ky Fan[4]” should be “Ky Fan [4]” (make this change in all the paper!!!) Line 10: “be the compact” should be “are compact” Line 11: “satisfied” should be “satisfies” Line 12: “hands” should be “hand” Line 16: “interested various types” should be “interested in various types” Line 16: “problems, they” should be “problems. They” Lines 17, 26, 44, 62, 76: “conditions theorems” should be “conditions” Line 18: “(see[1,2,8,9,16,17]).” should be “(see [1,2,8,9,16,17]).” Lines 22-23: “is not have” should be “doesn’t have” Line 23: “, we” should be “. In consequence, we” Line 28: “Huang[6] have constructed” should be “Huang [6] has constructed” Line 29: “problem, Huang” should be “problem. Huang” Line 32: “we are interesting” should be “we are interested” Line 40: “This paper has divided” should be “This paper is divided” Line 42: “Section 4 are our” should be “Section 4 include” Line 42: “in this paper, we” should be “in this paper: we” Line 43: “Mond-Weir type(2nd-MWD) and Wolfe type(2nd-WD)” should be “Mond-Weir type (2nd-MWD) and Wolfe type (2nd-WD)” Line 63: “And the complete proofs,” should be “For the complete proofs,” Line 69: “follows.” should be “follows:” Line 72: “problem (P), denote” should be “problem (P). Denote” Line 82: “such that” should be “it results” Lines 86, 88: “(strictly)Q” should be “(strictly) Q” Line 111: “problem (P),” should be “problem (P).” Line 112: “these dual models” should be “These dual models” Line 119, 196: “which satisfied the following” should be “satisfying” Lines 120, 197: “For … such that” should be “for … it follows” Line 124: “w.r.t. primary problem (P) are established as the following.” should be “with respect to primary problem (P) are established as follows.”

Reviewer 2 Report

In the paper the author consider a complex minmax fractional programming problem. In the study the author establish two dual models, namely the second-order Mond-Weir and Wolfe type models. Moreover, weak, strong and strictly converse duality theorems are proved. The problems considered in the paper are important, but the paper is written in a terrible way. There are many English language mistakes, which makes the reading of the paper very hard. Thus, the paper should be given to a substantial proofreading to eliminate those mistakes. In many places the author does not specify to which set the given variable/constant belongs, e.g., in Lemma 1 to which set the mu parameter belongs to? At the beginning of each case in the proof of Lemma 3 the author assume some inequality, but he does not give any comment from which fact we can infer this inequality. In the lemma we assume only that Re[Fi] is Theta-bonvex, but this assumption is used latter in the proof. In the proofs of theorems the author use Theorem 1, but in the proved theorems there is no assumption that the (P) problem satisfies the constraint qualification, which is required by Theorem 1. Without showing that the (P) problem satisfies this constrain we cannot use Theorem 1.